# Sialyl Lewis[x]-P-selectin cascade mediates tumor–mesothelial adhesion in ascitic fluid shear flow

Shan-Shan Li [1,10], Carman K.M. Ip[1,10], Matthew Y.H. Tang[2], Maggie K.S. Tang[1], Yin Tong [1], Jiangwen Zhang[1], Ayon Ahmed Hassan[1], Abby S.C. Mak[1], Susan Yung[3], Tak-Mao Chan[3], Philip P. Ip [4], Cheuk Lun Lee[5], Philip C.N. Chiu [5], Leo Tsz On Lee[6], Hung-Cheng Lai[7,8], Jin-Zhang Zeng[9], Ho Cheung Shum[2] & Alice S.T. Wong[1]

Organ-specific colonization suggests that specific cell–cell recognition is essential. Yet, very little is known about this particular interaction. Moreover, tumor cell lodgement requires binding under shear stress, but not static, conditions. Here, we successfully isolate the metastatic populations of cancer stem/tumor-initiating cells (M-CSCs). We show that the M-CSCs tether more and roll slower than the non-metastatic (NM)-CSCs, thus resulting in the preferential binding to the peritoneal mesothelium under ascitic fluid shear stress. Mechanistically, this interaction is mediated by P-selectin expressed by the peritoneal mesothelium. Insulin-like growth factor receptor-1 carrying an uncommon non-sulfated sialyl-Lewis[x] (sLe[x]) epitope serves as a distinct P-selectin binding determinant. Several glycosyl-transferases, particularly α1,3-fucosyltransferase with rate-limiting activity for sLe[x] synthesis, are highly expressed in M-CSCs. Tumor xenografts and clinical samples corroborate the relevance of these findings. These data advance our understanding on the molecular regulation of peritoneal metastasis and support the therapeutic potential of targeting the sLe[x]-P-selectin cascade.

[1] School of Biological Sciences, University of Hong Kong, Pokfulam, Hong Kong. [2] Department of Mechanical Engineering, University of Hong Kong, Pokfulam, Hong Kong. [3] Department of Medicine, University of Hong Kong, Queen Mary Hospital, Pokfulam, Hong Kong. [4] Department of Pathology, University of Hong Kong, Queen Mary Hospital, Pokfulam, Hong Kong. [5] Department of Obstetrics and Gynecology, University of Hong Kong, Queen Mary Hospital, Pokfulam, Hong Kong. [6] Centre of Reproduction Development and Aging, Faculty of Health Sciences, University of Macau, Taipa, Macau, China. [7] Department of Obstetrics and Gynecology, School of Medicine, College of Medicine, Taipei Medical University, Taipei 23561, Taiwan. [8] Department of Obstetrics and Gynecology, Shuang Ho Hospital, Taipei Medical University, Taipei 23561, Taiwan. [9] State Key Laboratory of Cellular Stress Biology and Fujian Provincial Key Laboratory of Innovative Drug Target Research, School of Pharmaceutical Sciences, Xiamen University, Xiamen 361102, China. [10]These authors contributed equally: Shan-Shan Li, Carman K.M. Ip. Correspondence and requests for materials should be addressed to H.C.S. (email: ashum@hku.hk) or to A.S.T.W. (email: awong1@hku.hk)

Ovarian cancer has the lowest 5-year survival rate ( < 25%) among all gynecological malignancies owing to extensive peritoneal metastatic lesions[1]. Despite not as common as blood-borne metastasis, treatment of peritoneal metastasis is notoriously challenging because of the rapid growth of metastasis in the peritoneal cavity in a positive feed-forward manner[2]. Current treatment is dissatisfying and little overall clinical benefits for patients have been achieved over the last several decades. Unraveling the underlying molecular mechanisms regulating this process will conceivably benefit the development of effective treatment strategies.

The barriers to metastasis are distinct in different organs, suggesting that specific recognition between cancer cells and peritoneal mesothelium is critical for the peritoneum-specific colonization[3]. Moreover, tumor cell lodgment in the peritoneum is consistently exposed to shear forces generated by ascitic flow[4]. However, owing to the difficulty in manipulating the dynamic flow conditions, the role of mechanical force in this adhesion has been largely neglected. Thus, the adhesion molecules and the underlying signaling operating the adhesion of cancer cells under ascitic flow remain a gap in knowledge.

Selectins (E-, P-, and L-selectin) are a family of calcium-dependent glycoproteins that are the prime glycan receptors on shear-resistant interactions as described to date[5]. Although the structures of the three selectins are highly similar, they have different tissue distribution and binding kinetics, suggesting that they have different roles in various pathophysiological processes, including tumor metastasis. Thus, the selective binding between selectin and its ligand determines the metastatic destination. Tumor cells exploit the selectin-glycan binding for the initial cell–cell interaction, including tethering and rolling, under shear stress that further triggers the molecular signaling that facilitates cellular firm adhesion[6]. Such interaction has been extensively studied in blood-borne metastasis[7,8]. In contrast, selectins are expressed on peritoneal mesothelial cells, suggesting that similar receptor/ligand cascade initiated by selectin-mediated interaction may promote tumor cell peritoneal targeting[9,10]. However, the detailed underlying mechanism governing adhesion for peritoneal metastasis is poorly understood. The difference in the peritoneal shear stress (~ 0.1 dyne cm$^{-2}$; at least 10-fold lower than that of the vascular shear stress)[11] suggests that the biology of peritoneal dissemination is different from that of blood-borne metastasis and probably different molecular mechanisms are involved.

In this study, we reveal a preferential binding of metastatic cancer stem/tumor-initiating cells (M-CSCs) to the peritoneal mesothelium than the non-metastatic (NM)-CSCs. We also provide evidence that P-selectin is a key molecule on the peritoneal mesothelium that mediates the binding through a unique sialyl-Lewis$^x$ (sLe$^x$) containing insulin-like growth factor receptor-1 (IGF-1R) ligand under ascitic flow-induced shear stress.

## Results

**Establishment of M-/NM-CSC models**. To explore the limiting factors that determine the metastatic success of ovarian cancer, we reasoned that metastatic colonization is a highly inefficient process and is accomplished by only a subset of cancer stem/tumor-initiating cells (CSCs)[3]. We have previously isolated CSCs[12]. Using in vivo selection by orthotopic implantation of CSCs into the bursa of the mouse ovary, we further isolated the highly metastatic population of CSCs (M-CSCs) in SKOV-3 (Fig. 1a), which consistently metastasized to the peritoneum, recapitulating the clinical progression of human ovarian cancer, when implanted either by orthotopic or intraperitoneal (i.p.) injection (Fig. 1b–e). Conversely, NM-CSCs, which are equally

tumorigenic (Fig. 1d, e), did not metastasize in both models (Fig. 1b–e, Table 1). Similar metastatic abilities were observed in CSCs-enriched cultures generated from primary ovarian cancer samples (Fig. 1f, Table 1). Moreover, M-CSCs had a gene expression profile similar to the metastatic tumors in ovarian cancer patients (Fig. 1g), suggesting that the in vitro M-CSCs could closely reflect the spontaneous metastatic cellular events in patients. We have also derived M- and NM-CSCs from HEYA8 cells, and similar results were obtained (Supplementary Fig. 1a, b).

**Differential capabilities in tethering, rolling, and adhesion**. To evaluate whether the distinct metastatic phenotypes in the two CSCs populations have different abilities in tumor–mesothelial interaction, we have developed a customizable microfluidic platform to overcome the technical limitations of the conventional approaches and to recapitulate peritoneal dissemination in dynamic flow[13]. CSCs spheroids were perfused to the microfluidic chip coated with primary human peritoneal mesothelial cells (HPMCs) under well-defined flow rate. As shown, the CSCs were effectively captured to HPMCs at 0.03–0.15 dynes cm$^{-2}$ (a physiologically relevant range of shear stress of the ascites[14]) with maximal activity at a shear stress of 0.05 dynes cm$^{-2}$ (Fig. 2a), consistent with a catch bond that requires a clear threshold shear to initiate rolling[15]. Strikingly, both the kinetic and mechanical properties that govern tumor–mesothelial interaction were significantly different between the two CSCs populations. Compared with NM-CSC, M-CSCs had a higher tethering frequency (1.8-fold at 0.05 dynes cm$^{-2}$) (Fig. 2a, Supplementary Fig. 1c) and lower rolling velocity (Fig. 2b, Supplementary Fig. 1d) on HPMCs, resulting in a higher percentage of firm adhesion of M-CSCs to HPMCs (2.3-fold at 0.05 dynes cm$^{-2}$) (Fig. 2c, Supplementary Fig. 1e). Although HPMCs is sufficient to induce epithelial-to-mesenchymal transition in certain contexts[16], we showed that these cells retained their epithelial morphology with closely packed colonies of cells (Supplementary Fig. 2a) and N-cadherin and vimentin mRNA levels were similar between the primary culture and the later subculture (Supplementary Fig. 2b).

**P-selectin mediates M-CSC-HPMC interaction under flow**. HPMCs clearly showed constitutive cell surface expression of selectins (Fig. 3a, b). We also performed real-time PCR expression to verify the expression of E-, P-, and L-selectin. Like endothelial cells, used as a reference cell population, selectins are expressed on HPMCs, suggesting that their expression outside the vasculature is present, which are consistent with previous observations[9,10] (Supplementary Fig. 2c). To determine which selectin on HPMCs plays a major role in capturing M-CSCs under flow, anti-selectin blocking antibodies were used. Treatment with anti-E-, P-, or L-selectin increased the rolling velocities of M-CSCs when compared with that of isotype control (24.8% increase by anti-E-selectin, 63.3% increase by anti-P-selectin, and 22.5% increase by anti-L-selectin in median rolling velocities) (Fig. 3c, Supplementary Fig. 3a). Notably, blockade of P-selectin almost completely abrogated (~ 88%) tumor–mesothelial adhesion at 0.05 dynes cm$^{-2}$, whereas inhibition of E-selectin or L-selectin had partial effects (48% by anti-E-selectin; 28% by anti-L-selectin) (Fig. 3d, Supplementary Fig. 6b, Supplementary Video 1, 2). Consistently, significant adherence to P-selectin-Fc (59%), but not E-selectin- or L-selectin-Fc, was observed with M-CSCs (Fig. 3e, Supplementary Fig. 3,c), confirming a predominant role for P-selectin. In contrast, NM-CSCs did not show similar interactions with selectin-Fcs. We further showed that CSCs in ovarian cancer patients' ascites utilized P-selectin-dependent binding as observed in M-CSCs, as tumor spheroids isolated from

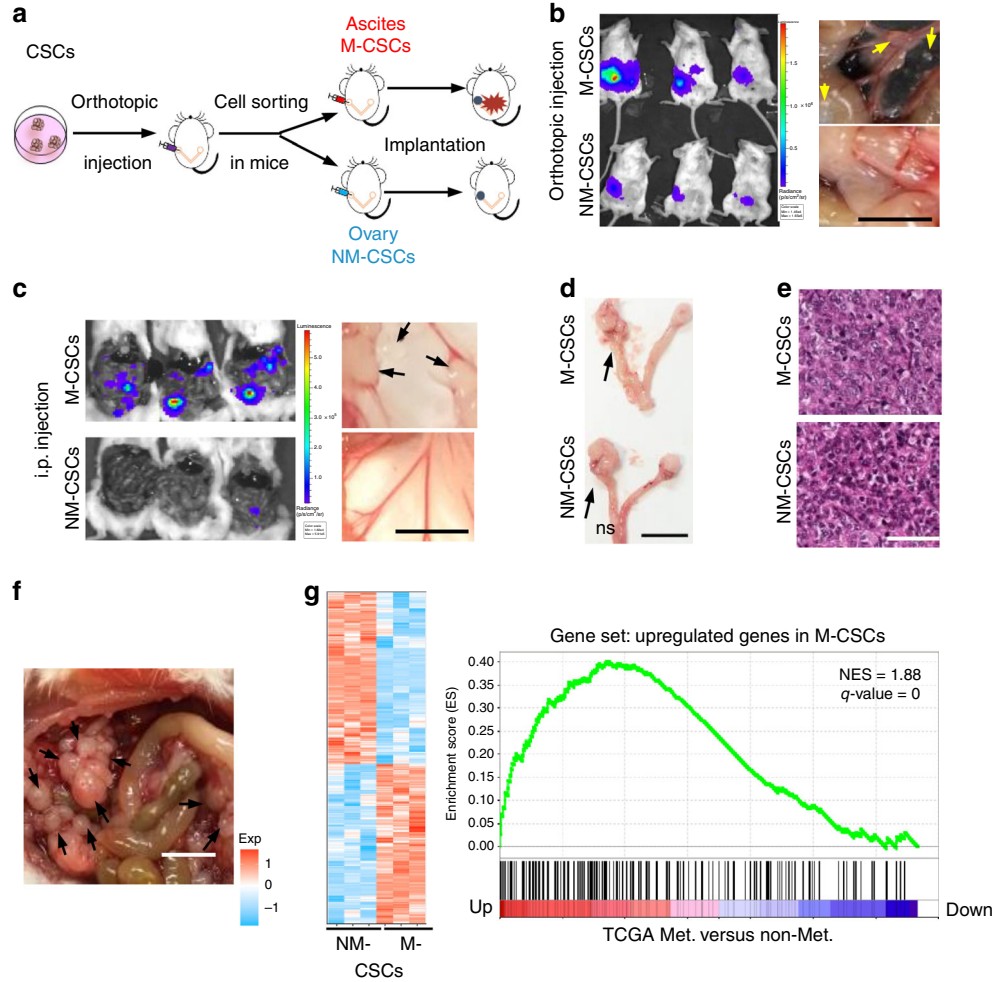

**Fig. 1** Isolation and characterization of metastatic cancer stem cells. **a** Isolation of ovarian metastatic cancer stem cells. Blue circle, primary tumor. Red star, metastasis. **b**, **c** NOD/SCID mice engrafted with M-CSCs or NM-CSCs via **b** orthotopic or **c** i.p. injection. Left: representative bioluminescence image. Right: representative views of metastases in the peritoneal cavity. Arrows, metastases. Scale bar, 1 cm. $n = 5$ (orthotopic) or three mice (i.p) per group. **d**, **e** Representative image **d** of and H&E staining **e** of primary tumors from mice orthotopically transplanted with M-CSCs or NM-CSCs. Arrows, tumors. Scale bars = 1 cm **d** or 50 μm **e**. **f** Representative view of metastasis in mice engrafted with M-CSCs of patient samples. Arrows, metastases. Scale bar, 1 cm. $n = 7$ mice, samples collected from three patients. **g** Gene expression patterns of M-CSCs and NM-CSCS compared with metastatic (Met.) versus non-metastatic (Non-Met.) tumors from TCGA RNA-seq data of ovarian cancer patients. Left: heat map representing genes in M-CSC and NM-CSCs. Rows represent different genes and column represents each sample. Blue: downregulation; Red: upregulation. Right: Gene Set Enrichment Analysis of differentially upregulated genes in M-CSCs samples against the ranked gene list (from up- to downregulated) in TCGA metastatic ovarian tumors. X axis from left to right: ordered genes from up- to downregulation in TCGA metastatic tumors. Experiments were conducted two **a**–**f** or three **g**, $n = 3$ per group) times independently. Results are represented with mean ± SEM. Statistical analysis using unpaired Student's $t$ test. ns, not significant. **, $P < 0.01$

**Table 1 Comparison of tumorigenic and metastatic abilities**

|  | NM-CSCs | M-CSCs | Patient samples |
|---|---|---|---|
| Tumorigenesis | Yes (5/5) | Yes (5/5) | Yes (7/7) |
| Metastasis | No (0/8) | Yes (8/8) | Yes (7/7) |
| Metastases (no.) | — | 39 ± 5 (orthotopic)[a], 23 ± 3 (i.p.)[a] | 13 ± 1 |
| Ascites volume (mL) | — | 0.5 ± 0.1 (orthotopic)[a], 0.3 ± 0.1 (i.p.)[a] | 3.65 ± 0.4 |

Results are represented with mean ± SEM from two independent experiments, unpaired Student's $t$ test. ns, not significant
[a]$P < 0.01$

ascites rolled and adhered onto the P-selectin-Fc but not Fc control (Fig. 3f, Supplementary Video 3, 4). The P-selectin-Fc-mediated binding was shear-resistant and $Ca^{2+}$-dependent. Once the M-CSCs had adhered to the HPMCs, the bonding between M-CSCs and HPMCs was strong that the M-CSCs can only be completely dissociated when a maximal shear stress of 4 dynes cm$^{-2}$ was applied (Supplementary Fig. 4a). Treatment of M-CSCs with ethylenediaminetetraacetic acid (EDTA) inhibited the adhesion of M-CSCs to HPMC (Supplementary Fig. 4b) and could easily detach the adherent M-CSCs from the HMPCs and P-selectin chimera, even at shear stress ~ 1 dynes cm$^{-2}$ (Supplementary Fig. 4c).

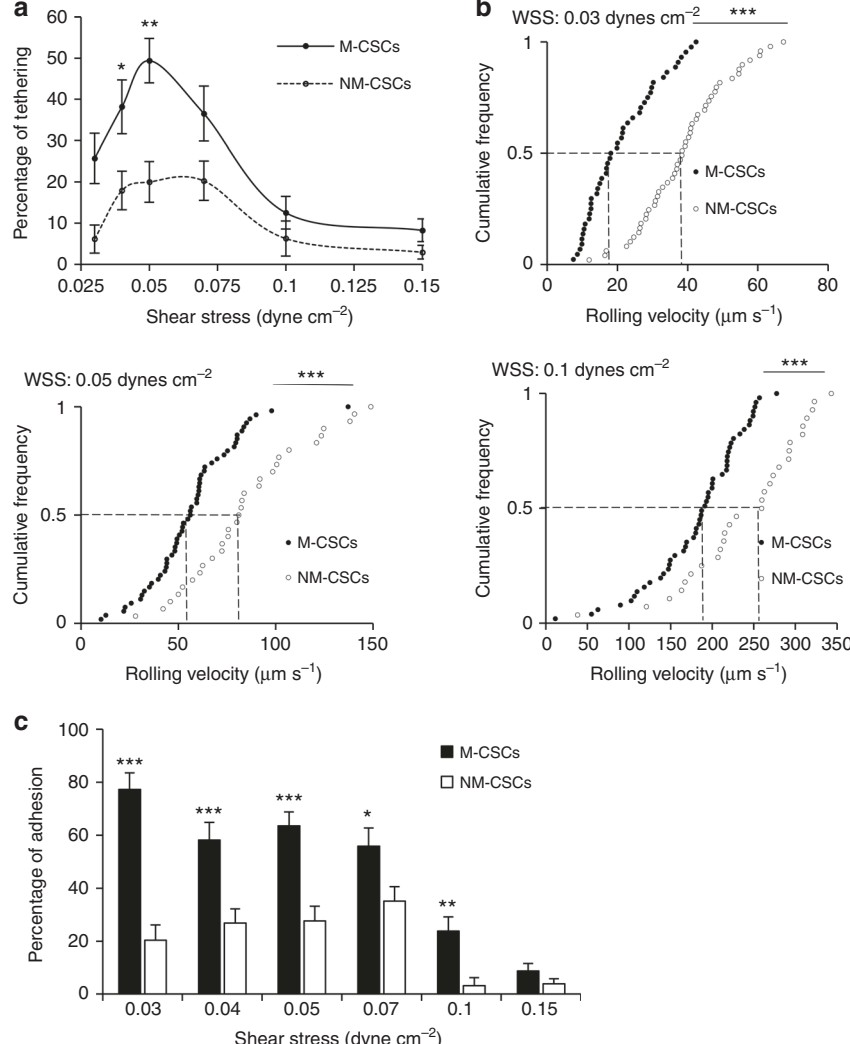

**Fig. 2** Differential capabilities of M- and NM-CSCs in rolling adhesion on HPMCs. **a** Percentage of tethering of M-CSCs and NM-CSCs on HPMCs. **b** CSCs rolling velocities on HPMCs monolayers. Each dot represents individual cancer spheroid. Dash lines: median velocity of two cell lines. **c** Percentage of adhesion of M-CSCs and NM-CSCs on HPMCs under different shear stress. Results are representative of three independent experiments. Error bars indicate SEM of the mean. $n = 44/49$ (0.03), 35/43 (0.04), 54/30 (0.05), 52/74 (0.07), 51/28 (0.1), 92/103 (0.15). Statistical analysis using chi-square test **a**, **c** and one-tailed unpaired Student's $t$ test **b**. ns, not significant. $*P < 0.05$; $**P < 0.01$; $***P < 0.001$

**P-selectin is required for peritoneal metastasis in vivo**. Next, we took advantage of the $Selp^{-/-}Rag2^{-/-}$ mice to study the requirement of P-selectin in intraperitoneal metastasis of M-CSCs. In mice with wild type P-selectin ($Selp^{WT}Rag2^{-/-}$), M-CSCs quickly disseminated throughout the peritoneal cavity after orthotopic or i.p. injection (Fig. 3g, h) with visible tumor nodules growing on the omentum, mesenteries and small bowels (Fig. 3g, h) and developed massive ascites (Supplementary Table 4). In contrast, the metastatic progression was markedly reduced in $Selp^{-/-}Rag2^{-/-}$ mice (Fig. 3g, h) with no significant difference in primary tumor growth (Fig. 3i, j). To further confirm the role of P-selectin and to explore whether therapeutic agent against P-selectin is able to eliminate tumor-mesothelium adhesion, we tested KF38789, a selective inhibitor of P-selectin-mediated cell–cell adhesion in vivo. Concurrent with the above results, KF38789 significantly inhibited the extent of tumor metastases on the omentum, mesenteries, and peritoneal wall when treated 1 h prior to the injection of M-CSCs derived from both SKOV-3 and ovarian cancer patients (Fig. 3k), confirming the specific role for P-selectin in mediating M-CSCs adhesion.

**sLe$^x$ on an O-glycoprotein is P-selectin ligand on M-CSCs**. Sialylation, fucosylation, and sulfation are the main features of P-selectin ligands[17]. As shown, the removal of sialic acid with neuraminidase reduced M-CSCs binding to HPMCs or P-selectin-Fc by ~ 50% (Fig. 4a). Notably, almost completely abolished M-CSCs binding to HPMCs or P-selectin-Fc was observed upon the removal of fucose with fucosidase (Fig. 4b), whereas inhibition of sulfation by sodium chlorate did not alter the binding activity (Fig. 4c), indicating that the binding between M-CSCs and HPMCs or P-selectin-Fc is highly fucosylation- and relatively weaker sialylation-, but not sulfation-dependent. We next examined the involvement of fucosylated and sialylated glycans, sLe$^x$ or sLe$^a$, in mediating M-CSCs adhesion. As shown in Fig. 4d, e, preincubation of M-CSCs with anti-sLe$^x$-blocking antibody reduced binding of M-CSCs to HPMCs and P-selectin-Fc, whereas anti-sLe$^a$-blocking antibody did not affect the interaction. By flow cytometry, whereas M-CSCs, but not NM-CSCs, displayed reactivity of HECA-452 and CSLEX-1, which recognizes sLe$^{a/x}$ and sLe$^x$, respectively, they showed no difference on anti-sLe$^a$ (CA19-9) expression (Fig. 4f, Supplementary Fig. 5). We further found that sLe$^x$-containing glycan is modified on a

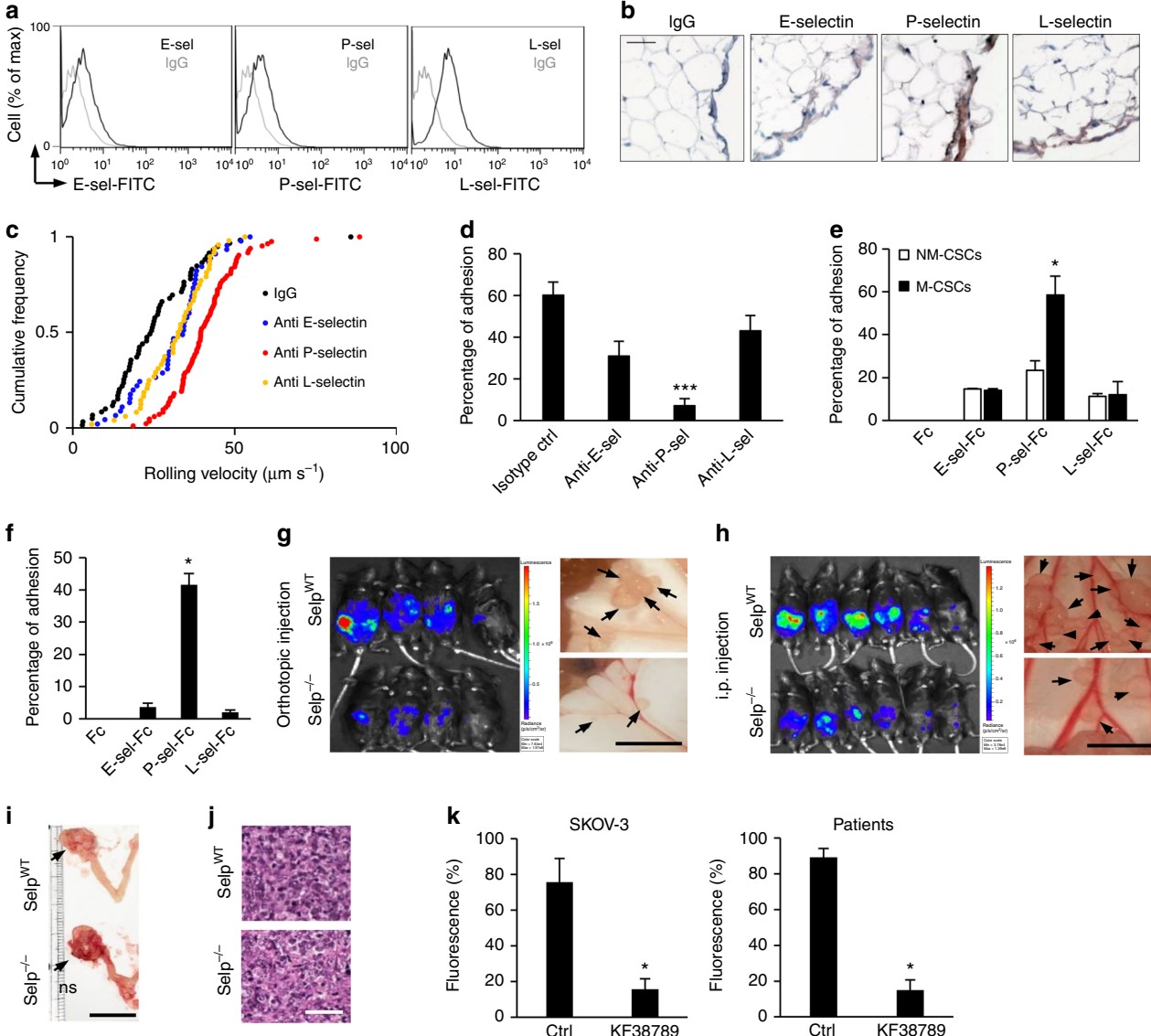

**Fig. 3** P-selectin regulates metastatic progression of M-CSCs. **a** Cell surface expression of selectins on HPMCs. E-selectin (E-sel), P-selectin (P-sel), L-selectin (L-sel). **b** Human omentum tissues stained with anti-selectin antibodies. Scale bar, 50 μm. **c, d** Rolling velocity **c**, adhesion percentage **d** of M-CSCs on HPMCs pre-incubated with anti-selectins antibodies or IgG at 0.05 dynes $cm^{-2}$. $n = 59, 45, 83, 48$. **e** Percentage of CSCs adhesion onto selectin recombinant proteins or Fc at 0.05 dynes $cm^{-2}$. $n = 35/39, 46/45, 52/63, 54/52$. **f** Percentage of patient ascites-derived tumors adhesion onto selectins at 0.05 dynes $cm^{-2}$. $n = 100, 131, 160, 247$, tumor samples collected from three patients. Data (mean ± SEM) from one of three independent experiments, chi-square test **d-f**. **g, h** P-selectin wild-type ($Selp^{WT}$) or knockout ($Selp^{-/-}$) $Rag2$ deficiency mice orthotopically **g** or i.p. **h** inoculated with M-CSCs cells. Arrows: metastases. Scale bar, 1 cm. $n = 6$ mice (orthotopic) or nine mice (i.p.) from two independent experiments. **i, j** Representative image **i** and H&E staining **i** of primary tumors in the ovary. Arrows, tumors. Scale bars = 1 cm **i** or 50 μm **j**. **k** Fluorescent signal of tumor cells adherent on mouse peritoneum. M-CSCs derived from SKOV-3 (left, $n = 3$ mice) and patient samples (right, $n = 6$ mice, tumor samples collected from two patients). Data (mean ± SEM) from two independent experiments, unpaired Student's $t$ test **i, k**. ns, not significant. *$P < 0.05$; **$P < 0.01$

protein but not a lipid, as protease (trypsin) treatment on M-CSCs almost completely abolished the adhesion of M-CSCs on both HPMCs and P-selectin-Fc (Fig. 4g), whereas glycolipid synthesis inhibitor (PPMP) treatment had no effect (Fig. 4h). Treatment of O-sialoglycoprotein endopeptidase (OSGE) substantially reduced the adhesion of M-CSCs on both HPMCs and P-selectin-Fc (Fig. 4i), whereas treatment of Peptide:N-glycosidase F (PNGase) for removal of N-glycans had no effect (Fig. 4j).

**P-selectin ligand on M-CSCs is mainly conferred by IGF-1R.** To identify the protein presenting the $sLe^x$ on M-CSCs that mediates the P-selectin interaction on HPMCs, we examined a set

of inhibitors at effective concentrations that specifically target various cell surface receptor tyrosine kinase (RTK), including c-Met, EGFR, FGFR, and IGF-1R, which have been previously implicated in the progression of ovarian cancer[18–21]. We found that only genistein (general RTK inhibitor) and AG1024 (IGF-1R inhibitor) led to a marked decrease in the binding of M-CSCs to HPMCs or P-selectin-Fc (Fig. 5a, Supplementary Fig. 6). Moreover, blockade of M-CSCs adhesion on HPMCs or P-selectin-Fc binding was observed with the anti-IGF-1R (Fig. 5b). Furthermore, endogenous IGF-1R could be readily detected in the P-selectin-Fc immunoprecipitate (Fig. 5c). Significant amount of $sLe^x$ was detected in both IGF-1R subunits (Fig. 5d). The levels of

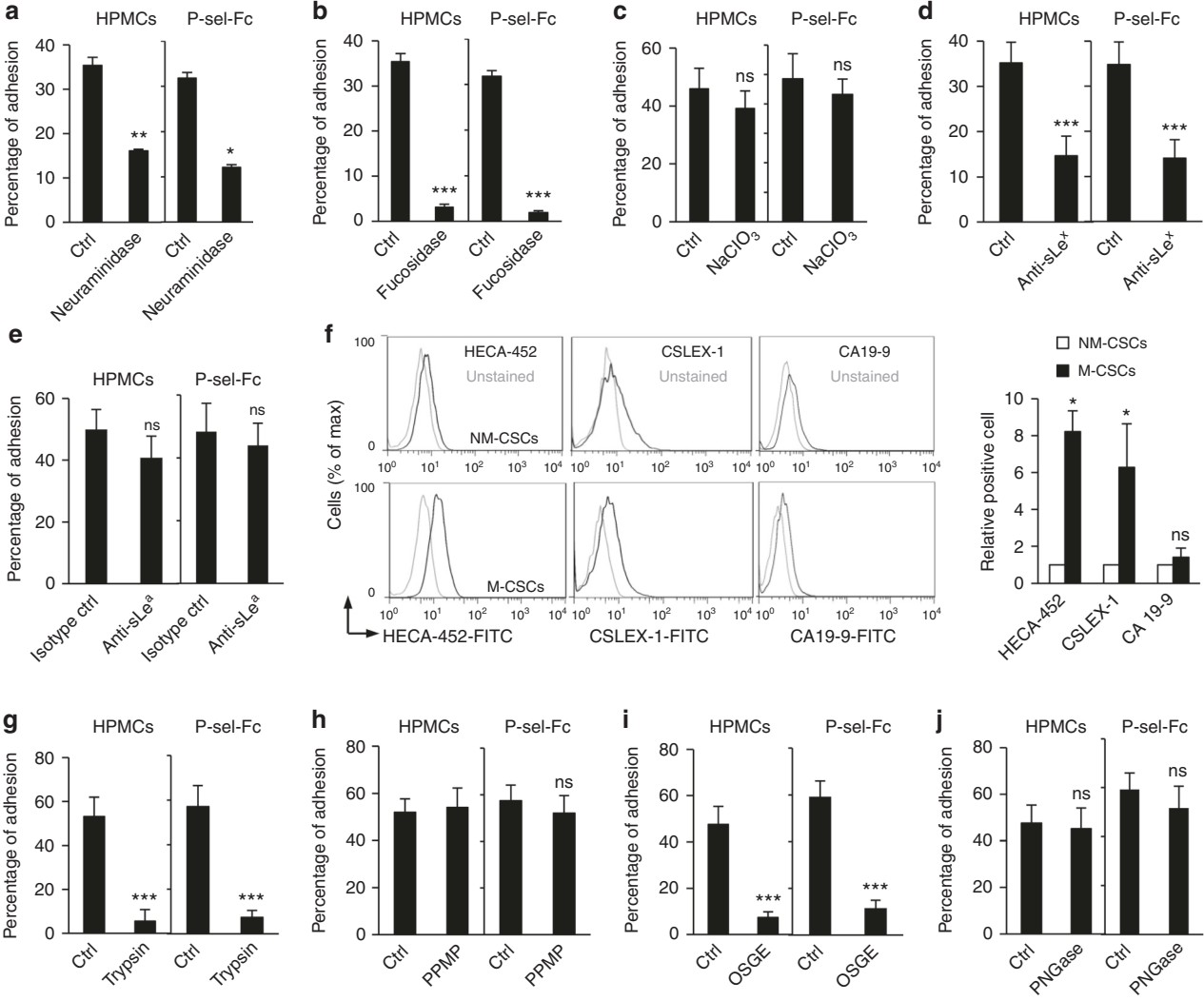

**Fig. 4** sLe$^x$ mediates M-CSCs interaction with P-selectin on HPMCs. **a–e** Percentage of M-CSCs adhesion to HPMCs or P-selectin-Fc after treatment. **a** Neuraminidase treatment. $n = 73/35$, $46/108$. **b** Fucosidase treatment. $n = 98/153$, $67/64$. **c** NaClO$_3$ treatment. $n = 49/65$, $35/89$. **d** Anti-sLe$^x$ treatment. $n = 108/68$, $89/71$. **e** Anti-sLe$^a$ treatment. $n = 108/68$, $89/71$. **f** Cell surface expression of sLe$^{a/x}$ on CSCs. Left: representative images of flow cytometry. Right: relative positive cell plot. $n = 3$. **g–j** Percentage of M-CSCs adhered to HPMCs or P-selectin-Fc after treatment. **g** Trypsin treatment. $n = 32/36$; $28/68$. **h** PPMP treatment. $n = 77/37$, $54/46$. **i** OSGE treatment. $n = 42/109$, $54/95$. **j** PNGase treatment. $n = 42/31$, $54/31$. Data (mean ± SEM) from one of three independent experiments, chi-square test **a–e**, **g–j**, or from three biological replicates, unpaired Student $t$ test **f**. ns, not significant. *$P < 0.05$; **$P < 0.01$

phospho-IGF-1R were elevated soon upon P-selectin-Fc binding (Fig. 5e). These experiments were all performed in the absence of IGF-1, suggesting a ligand-independent activation of IGF-1R upon P-selectin binding. The known P-selectin ligands bearing sLe$^x$, such as CD24, was detected. However, unlike IGF-1R, there was no differential expression of CD24 between M-CSCs and NM-CSCs (Fig. 5f).

**FUT5 is critical for ovarian tumor progression.** To further define the molecular properties that contribute to P-selectin binding, we examined the expression profile of glycosyl-transferases involved in sLe$^x$ synthesis (Fig. 6a) in M-CSCs and NM-CSCs. We found that several glycogenes, including *B4GalT4*, *ST3Gal3*, *ST3Gal4*, and *FUT5*, were significantly increased in M-CSCs when compared with NM-CSCs (Fig. 6a, Supplementary Fig. 7a). Similar higher expression of these genes on patients' derived CSCs was also observed (Fig. 6a). The high mRNA expression of glycosyltransferases correlates with poorer progression-free survival in patients with advanced (stage III, IV)

ovarian cancer (Supplementary Fig. 7b). Among these genes, we are particularly interested in *FUT5*, which encodes a rate-limiting enzyme α1,3-fucosyltransferase catalyzing the addition of fucose residue for sLe$^x$ synthesis[22,23]. *FUT5* knockdown abolished sLe$^x$ cell surface expression on M-CSCs (Fig. 6b) and largely reduced the adhesion of M-CSCs to HPMCs and P-selectin-Fc under shear stress (Fig. 6c). Moreover, mice inoculated with orthotopic or i.p. xenograft (Fig. 6d) of M-CSCs carrying *FUT5* shRNA had significantly reduced metastatic implants and ascites formation in the peritoneal cavity when compared with mice injected with nonspecific (NS) shRNA M-CSCs (Supplementary Table 5), with no apparent difference in primary tumor growth (Fig. 6e, f). Real-time PCR of tumor samples harvested from mice at the end of the study confirmed successful target gene knockdown of *FUT5* (Fig. 6g).

## Discussion

Tumor microenvironment is one of the major factors controlling the metastatic progression; however, the detailed molecular

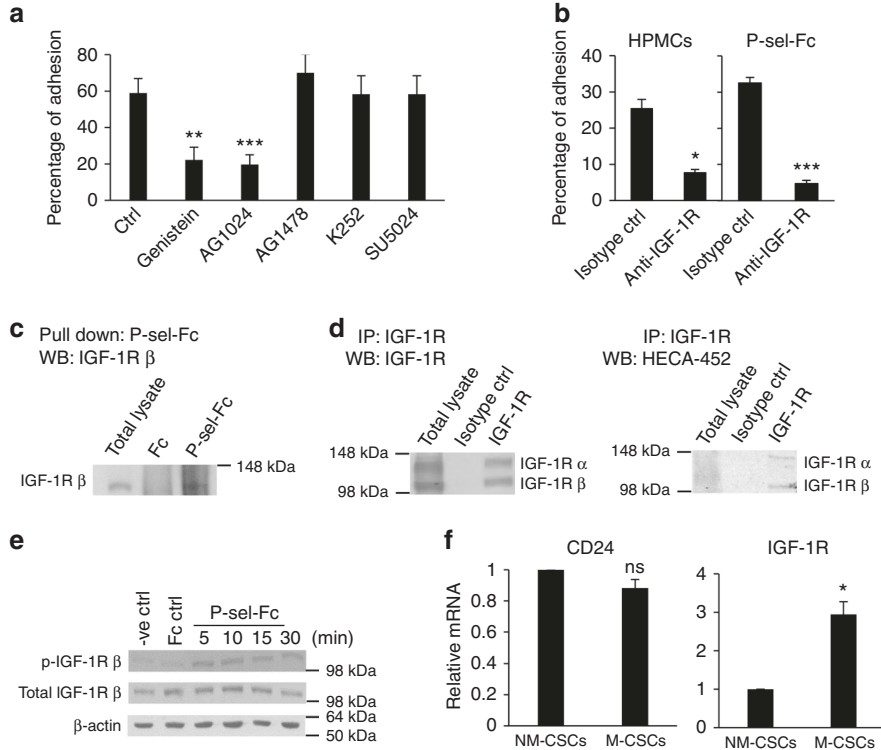

**Fig. 5** The sLe$^x$-containing P-selectin ligand is IGF-1R-dependent. **a**, **b** Percentage of adherent M-CSCs after treatment with RTK inhibitors on HPMCs. Genistein, general RTK inhibitor; AG1024, IGF-1R inhibitor; AG1478, EGFR inhibitor; K252a, c-Met inhibitor; SU5402, FGF-1R inhibitor. $n = 39, 36, 56, 20, 24, 24$. **b** Percentage of M-CSCs adhesion to HPMCs or P-selectin-Fc after anti-IGF-1R treatment. $n = 45/71, 70/72$. Data (mean ± SEM) from one of three independent experiments, chi-square test **a**, **b**. **c** Presence of IGF-1R in the M-CSCs protein lysate pulled down with P-selectin-Fc. **d** Detection of HECA-452 antigen on IGF-1R of M-CSCs. **e** P-selectin binding on M-CSCs activates the phosphorylation of IGF-1R. **f** mRNA expression of *CD24* (left) and *IGF-1R* (right) in M-CSCs and NM-CSCs. Data (mean ± SEM) from three biological replicates, $n = 3$, unpaired Student $t$ test. ns, not significant; *$P < 0.05$; **$P < 0.01$; ***$P < 0.001$

mechanisms that operate the dissemination of ovarian cancer within the peritoneal cavity, particularly under ascitic shear stress, have not been explored. In this study, we have provided evidence that a sLe$^x$-bearing glycan on IGF-1R of ovarian M-CSCs interacts with P-selectin on HPMCs controls the tethering, rolling, and subsequent adhesion in peritoneal dissemination under shear stress.

Several lines of evidence suggest that CSCs may be a key player in the metastasis of ovarian cancer. First, ovarian CSCs are enriched in cells that can undergo epithelial–mesenchymal transition, a key process in metastasis[24]. Second, CSCs have higher tumorigenicity[25]. Third, ovarian cancer cells possess a stem-like gene signature correlates with poor progression-free and overall survival in cancer patients[26]. Fourth, the existence of these CSCs at an early stage of ovarian cancer is consistent with the frequent clinical observation of early metastasis[2]. In recent years, several studies in various cancer types, including breast and colorectal cancer, showed only a subpopulation of CSCs are able to metastasize[27–29]. In this report, we showed that the highly metastatic subpopulation CSCs in ovarian cancer ascites; and more importantly, we provided a detailed molecular understanding on their metastatic ability through a direct interaction with the peritoneal microenvironment, which not been studied in previous metastatic CSC models.

Very little is known about the selectin ligands on ovarian carcinoma cells[30]. Our present work has uncovered a sLe$^x$-bearing P-selectin ligand in ovarian cancer cells, which is distinct from those previously defined P-selectin ligands, in particular the well-known ligand PSGL-1, which bears tyrosine sulfation. Much evidence hinted the casual relationship between sLe$^x$ and ovarian

cancer metastasis. For example, antibodies against sLe$^x$ epitope have been previously shown to react with human ovarian cancer cells[31]. Glycosyltransferases that catalyze the synthesis of sLe$^x$ are significantly elevated in ovarian carcinoma tissues and cell lines[32,33]. High sLe$^x$ expression is associated with poor survival of patients with ovarian cancer[34]. The sLe$^x$-bearing glycan present on M-CSCs binds P-selectin in a rigid fucosylation-dependent and relatively loose sialylation-dependent manner, instead of a simple sLe$^x$ structure that relies equally on fucose and sialic acid moieties[35]. Such biochemistry is of biological significance, as fucosylation has been shown with increased affinity and bind selectin more efficiently[36–38]. Whereas sulfation is a common feature of P-selectin ligands[39,40], our results suggest that the sLe$^x$-bearing glycan on M-CSCs does not require sulfation for its binding to P-selectin. There is evidence that sulfated moieties of P-selectin ligands tend to be more resilient to stress for binding under high shear conditions[41].

Although consistent with a direct role for P-selectin in mediating the tumor–mesothelial interaction, our findings also appear to differ somewhat from those of a recent work to show the sLe$^x$-bearing P-selectin ligand[10]; however, the use of cells with different genetic backgrounds in this recent work may explain the difference. While CD24 expression is higher in some cell lines, the cell adhesion via P-selectin seems less, suggesting that there is other sLe$^x$-containing ligand(s) of P-selectin. Using carefully controlled conditions and in comparison with a non-metastatic counterpart, while we have also shown CD24 expression in M-CSCs and NM-CSCs, there was no differential expression on CD24, unlike IGF-1R, in M-CSCs and NM-CSCs, suggesting the importance of IGF-1R in mediating the adhesion of M-CSCs. Moreover, our findings

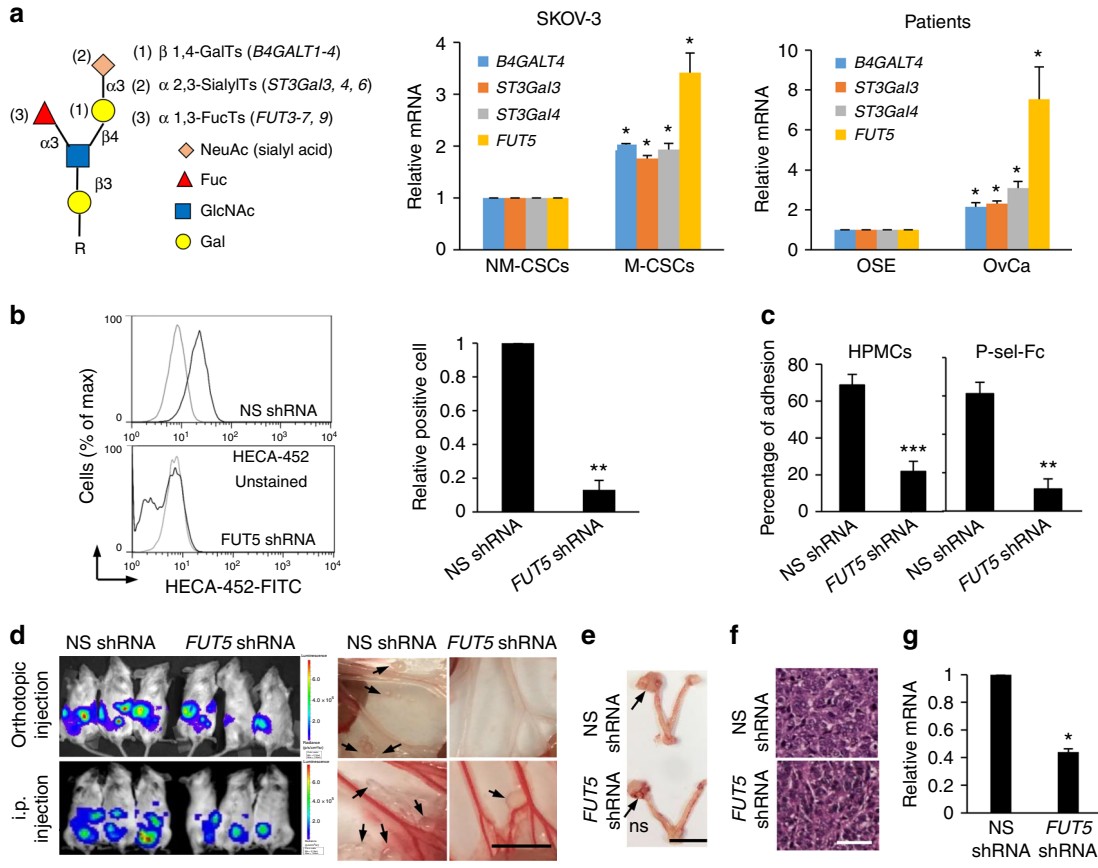

**Fig. 6** *FUT5* is critical for ovarian cancer progression. **a** mRNA expression of glycogenes related with sLe$^x$ biosynthesis. Left: schematic image indicating sLe$^x$ biosynthesis. sLe$^x$ is synthesized by sequential addition of N-acetylglucosamine (GalNAc), galactose (Gal), sialic acid (NeuAc), and fucose (Fuc) to the backbone catalyzed by N-acetylglucosaminyltransferases (GnTs), β1,4-Galactosyltransferase (*B4GalT1-4*), α 2,3-Sialyltransferases (*ST3Gal3, 4, 6*) and α 1,3-Fucosyltransferase (*FUT3-7, 9*). Middle: mRNA expression of glycogenes in M-CSCs and NM-CSCs. $n = 3$ per group. Right: mRNA expression of glycogenes in normal human ovarian surface epithelial (OSE) cells and M-CSCs collected from ovarian cancer patients (OvCa). $n = 2$ patients (OSE) or $n = 4$ (OvCa). Data represent as mean ± SEM, unpaired Student's *t* test. **b** Detection of HECA-452 antigen on *FUT5* knockdown M-CSCs. Data (mean ± SEM) from three biological replicates, $n = 3$, unpaired Student's *t* test. **c** Percentage of *FUT5* knockdown M-CSCs adhered onto HPMCs or P-selectin-Fc. $n = 64/55$, $67/35$. Data (mean ± SEM) from one of three independent experiments, chi-square test. **d** Metastasis in mice with NS or *FUT5* shRNA transduced M-CSCs orthotopic (upper) or i.p. (lower) xenograft model. Arrows: metastases. Scale bar, 1 cm. $n = 3$ mice (orthotopic) or six mice (i.p.). **e, f** Representative images **e** and H&E staining **f** of primary tumors in mice. Arrows, tumors. Scale bars = 1 cm **e** or 50 μm **f**. **g** *FUT5* mRNA expression of in primary tumors. $n = 3$ mice per group. Data (mean ± SEM) from three biological replicates, $n = 3$, unpaired Student's *t* test. Experiments were conducted two **d–f** or three **a–c**, **g** times independently. ns, not significant. *$P < 0.05$. **$P < 0.01$; ***$P < 0.001$

are in agreement with a large literature that links IGF-1R with tumor progression.

The sLe$^x$-bearing P-selectin ligand is presented on IGF-1R, which is frequently overexpressed in ovarian cancer and other peritoneal metastasis models and confers a poor clinical prognosis[42,43]. Recently, sLe$^x$ has been identified to be decorated on various unexpected proteins, and functions of the sLe$^x$ are largely unknown[44]. IGF-1R, which to the best of our knowledge, has not been shown previously to possess sLe$^x$. Although incompletely understood, there is growing appreciation for a role of sLe$^x$ in modulating protein activity. In gastric cancer cells, increased sLe$^x$ of c-Met was associated with increased dimerization and phosphorylation, which resulted in increased c-Met-mediated signaling associated with tumor invasiveness[45]. Although we surmise that IGF-1R is functionally relevant for ovarian cancer progression and metastasis, our work suggests that blocking the sLe$^x$ mediating IGF-1R will be required to provide a promising effect.

In addition, our findings underscore a critical role for FUT5 during the development and maintenance of the metastasis. Higher incidence of sLe$^x$ in M-CSCs coincided with higher expression of glycosyltransferases. Ovarian cancer patients with *FUT5*-over-expressing tumors correlates with poorer survival compared with patients with low or no *FUT5*-expressing tumors. Importantly, blocking *FUT5* expression in M-CSCs serves as an effective strategy for the treatment of peritoneal dissemination. Most of the glyco-syltransferases genes involved in sLe$^x$ synthesis are constitutively expressed to produce sLe$^x$ direct precursor, whereas, *FUTs* encoding α1,3-fucosyltransferases catalyzing the last and rate-limiting step of sLe$^x$ synthesis by adding fucose to the precursor are normally switched off[22,23]. Moreover, peritoneal colonization of gastric cancer cells was reported to be suppressed by the downregulation of *FUT5*[46]. These data suggest that it is the context in which the glycan is expressed contributes to organotropism.

In summary, this study provides evidence showing P-selectin as a key molecule on HPMCs and IGF-1R carrying sLe$^x$ on M-CSCs as a ligand in mediating peritoneal dissemination under ascitic flow-induced shear stress (Fig. 7). This research is not only relevant to ovarian cancer, but also applicable to other tumor types, such as breast and colon cancers, in which peritoneal metastasis is an important pathological process, indicating that sLe$^x$-P-selectin could become promising therapeutic targets.

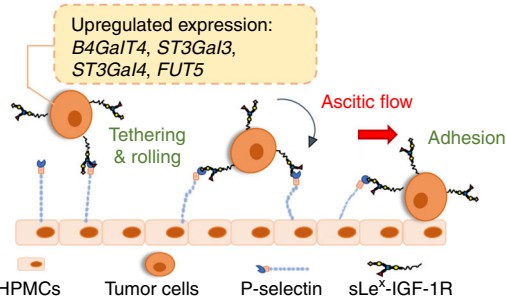

**Fig. 7** Schematic of sLe$^x$-P-selectin binding in peritoneal metastasis

## Methods

**Cells and cell culture.** Human ovarian cancer cell lines SKOV-3 and HEYA8 (kind gifts from Dr. N. Auersperg at University of British Columbia and Dr. J. Liu at MD Anderson Cancer Center, respectively) were maintained in Medium 199:MCDB105 containing 5% fetal bovine serum (FBS). CSCs were enriched from luciferase-labeled SKOV-3 and HEYA8 through several generations of serum-free low adherent culture and stem-like properties were characterized as described[12]. HEK293 cells (CRL-1573, ATCC) were cultured in Dulbecco's Modified Eagle Medium containing 10% FBS. Human umbilical vein endothelial cells (CC-2519, Clonetics) were cultured in F12K supplemented with 10% FBS, 20 µg mL$^{-1}$ endothelial cells growth supplement, 90 U mL$^{-1}$ heparin and 1% penicillin and streptavidin. Primary tumor samples were obtained from ovarian cancer patients with informed consent and approval by the Taipei Medical University Institutional Review Board and isolated as described[25]. The use of human tumor surgical specimens and tissue samples was approved by the Institutional Ethical Review Board at the University of Hong Kong. Normal human ovarian surface epithelial (OSE) cells were derived from surface scrapings of normal ovaries from women with nonmalignant gynecological diseases. HPMCs isolated from dialysate effluent from peritoneal dialysis from patients with nonmalignant disorders[47] were maintained in Medium 199:MCDB105 supplemented with 10% FBS, 1% penicillin and streptavidin. HPMCs within three passages were used to ensure genetic stability of the culture.

**Isolation of M-CSCs.** CSCs ($1 \times 10^5$) were injected into the ovary of NOD/SCID mouse (female, 6–8 weeks of age) to allow the formation of primary tumor and intraperitoneal dissemination. M-CSCs were collected from the ascites, whereas NM-CSCs were collected from the tumor burden remained in the ovary at 3 weeks' post inoculation. M-CSCs or NM-CSCs were maintained in serum-free medium in low adherent culture dishes as early described. To further verify the tumorigenesis and metastasis potential of the isolated populations, NM-CSCs, or M-CSCs were orthotopically ($1 \times 10^5$) or intraperitoneally ($5 \times 10^5$) injected into NOD/SCID mice. Tumor burden was monitored by bioluminescence imaging and mice were harvested at 3–4 weeks post inoculation. All animal experiment protocols were approved by the Committee on the Use of Live Animals in Teaching and Research at the University of Hong Kong.

**Antibodies.** The antibodies or recombinant proteins used in the flow cytometry, western blot, immunohistochemistry, or microfluidic perfusion assays are detailed in the Supplementary Table 1.

**RNA sequencing and TCGA data analysis.** Total RNA extracted from three pairs of M-CSCs and NM-CSCs were sent to Beijing Genomics Institute for RNA sequencing (RNA-seq). Differential gene expression between M-CSCs and NM-CSCs was determined by a moderated $t$ test with limma software. In total, 200 genes were identified upregulated in the M-CSCs compared with NM-CSCs ($\log_2$(fold-change) > 1 and $P < 0.05$), and 216 genes were identified downregulated ($\log_2$(fold-change) < −1 and $P < 0.05$). Hierarchical clustering was conducted on expression profile of 416 differentially expressed genes (DEG) and a high degree of internal consistency was discovered within each sample group. To discover whether M-CSCs could reflect the clinical metastatic process, we computed the fold-change of these genes in metastatic and non-metastatic tumors from The Cancer Genome Atlas (TCGA) ovarian cancer RNA-seq data. To quantify the consistency, we conduct the Gene Set Enrichment Analysis of DEG in M-CSCs samples against the ranked gene list (from up- to downregulated) in TCGA ovarian metastatic tumors.

**Flow cytometry.** Cultured adherent cells or tumor spheroids were detached with 2 mM EDTA PBS, suspended in PBS and counted. Cells were incubated with primary antibodies, selectin-Fc chimeras (5 µg mL$^{-1}$ in 1 mM CaCl$_2$, 1 mM MgCl$_2$, PBS-binding buffer) or isotype control per the manufacturer's instructions. The primary antibody or recombinant proteins were stained with appropriate Alexa Fluor 488 secondary antibodies. Events were collected on FACS AriaIII (BD Biosciences). Data were analyzed using FlowJo software (Tree Star Inc.) with gating strategy shown in Supplementary Fig. 8.

**Microfluidics CSCs perfusion assay.** Microfluidic chips were fabricated as described[13]. To coat microfluidic channels with selectin recombinant proteins, channels were incubated with 1 µg mL$^{-1}$ selectin-Fc recombinant proteins in PBS at 4 °C overnight. To coat microfluidic channels with HPMCs, channels were incubated with 10 µg mL$^{-1}$ human fibronectin in serum-free medium overnight at 4 °C. After wash, the channels were introduced with HPMCs suspension ($3.5 \times 10^6$ mL$^{-1}$) and cultured overnight under CO$_2$ incubator at 37 °C. To functionally block selectins, HPMCs were incubated with 20 µg mL$^{-1}$ anti-E-selectin, anti-P-selectin, anti-L-selectin or isotype antibody at room temperature for 1 h before the assays.

CSCs of 70–100 µm diameter were collected with cell strainer, fluorescently labeled with CMFDA Celltracker (2.5 µg mL$^{-1}$, C7025, Life Technologies) as per manufacturer's instructions for assays. Fluorescently labeled CSCs resuspended in binding buffer (1500 spheroids mL$^{-1}$) were perfused into the microfluidic channel through a syringe pump (LongerPump) under desired shear stresses. The viscosity of the binding buffer (0.9 cp) is within the range of viscosity (0.95 ± 0.15 cp) of ascitic fluid in patients with various ascitic etiologies[48]. The motion of cancer spheroids was observed under fluorescent microscope (Nikon ECLIPSE Ti) and serial time lapse images (1 image sec$^{-1}$) were captured for 30 s. More than three randomly chosen areas (× 4 objective) were captured for each treatment.

The motion of cancer spheroids under flow was analyzed offline using Image J (National Institutes of Health) and used for the classification of interactions as described[49]. In brief, cancer spheroids that tethered to the surface, then detached in the flow were defined as tethering. Cancer spheroids that moved at a velocity below the hydrodynamic velocity for more than one spheroid diameter were defined as rolling. Cancer spheroids that remained stationary for more than one frame during recording were determined as adhesion. Percentage of tethering or adhesion was calculated by dividing the number of tethering or adherent spheroids by the total number of spheroids flew through the field of observation in the 30 s duration.

In some experiments, M-CSCs were incubated with or without neuramidase (100 mU, N2876, Sigma-Aldrich), α1,3/4-fucosidase (150 mU, P0769S, New England BioLabs), OSGE (100 µg mL$^{-1}$, CLE100, Cedarlane) or PNGase (10 U mL$^{-1}$, P0704S, New England BioLabs) according to the manufacturer's instructions for 1 h at 37 °C prior staining. To inhibit sulfation, M-CSCs were incubated with sodium chlorate (25 mM, 403016, Sigma-Aldrich) in serum-free medium for 48 h. To functionally block sLe$^x$, sLe$^a$ or IGF-1R on M-CSCs, M-CSCs were incubated with CSLEX-1 (10 µg mL$^{-1}$, 551344, BD Biosciences), anti-CA19–9(10 µg mL$^{-1}$, ab15146, Abcam), anti-IGF-1R (2 µg mL$^{-1}$, 24–60, Invitrogen) or isotype antibodies for 1 h at 37 °C, respectively. To remove cell surface glycoprotein, M-CSCs were incubated with Trypsin/EDTA solution (0.05%, 25300054, Gibco) for 5 min and neutralized with the complete medium. To inhibit glycolipid synthesis, M-CSCs were cultured with or without DL-PPMP (2.5 µM, sc-205655, Santa Cruz) in serum-free medium for 5 days. To test the involvement of RTKs in mediating CSCs adhesion, M-CSCs were treated with or without small molecule inhibitors as indicated in the Supplementary Table 2 before perfusion. All CSC perfusion experiments were repeated with HPMCs isolated from at least three different patients.

**Adhesion of tumor cells to the mouse peritoneum.** M-CSCs ($1 \times 10^7$) derived from SKOV-3 and ovarian cancer patients were fluorescently labeled with CMFDA and i.p. injected into NOD/SCID mice. To inhibit P-selectin, KF38789 (1 mg kg$^{-1}$, 2748, Tocris) or vehicle control were injected into the peritoneal cavity of mice 1 h prior to cell injection. Mice were killed after 16 h and the peritoneal cavity was washed with PBS to remove non-adherent cells. Fluorescent signal from adherent tumor cells was acquired in the IVIS Spectrum and peritoneum, mesentery, and omentum were lysed with 1% NP-40 and fluorescence was measured with VICTOR (PerkinElmer).

**P-selectin knockout Rag2 immunodeficiency mice.** C57BL/6 Selp$^{-/-}$ (JAX #002289, The Jackson Laboratory) was crossbred with Rag2$^{-/-}$ (JAX #008449, The Jackson Laboratory) mice to generate Selp$^{+/-}$ Rag2$^{+/-}$ heterozygous offspring. The heterozygous offspring were further crossbred and to generate homozygous P-selectin wild type (Selp$^{WT}$ Rag2$^{-/-}$) and P-selectin knockout (Selp$^{-/-}$ Rag2$^{-/-}$) mice that were verified and selected with genotyping in the Jackson Lab. PCR genotyping of the Selp and Rag2 mutant was conducted with primers described in Supplementary Table 3. The homozygous strains were maintained by breeding between the mice with the same genotypes respectively. SKOV-3 M-CSCs were orthotopically inoculated ($3 \times 10^6$) or i.p. injected ($5 \times 10^6$) into Selp$^{WT}$ Rag2$^{-/-}$ or Selp$^{-/-}$ Rag2$^{-/-}$ mice (female, 6–8 weeks of age) and tumor metastatic progression was monitored as described.

**Western blot.** Proteins (20 µg) were resolved on sodium dodecyl sulphate–polyacrylamide gels and transferred to the nitrocellulose membrane (Bio-Rad). Target proteins were detected using specific primary antibodies overnight at 4 °C. The primary antibodies were detected with appropriate horseradish peroxidase conjugated secondary antibodies and visualized with enhanced chemiluminescence (PerkinElmer). Scans of the full blots are shown in Supplementary Fig. 9.

**Pull down assay.** M-CSCs were lysed in ice-cold lysis buffer (50 mM Tris pH 7.4, 150 mM NaCl, 1 mM CaCl$_2$, and 1% Triton X-100, and protease inhibitors cocktail (phosphatase inhibitors 1 mM Na$_3$VO$_4$, 1 mM NaF and protease inhibitors 1 µg mL$^{-1}$ pepstatin, 2 µg mL$^{-1}$ leupeptin, 4 µg mL$^{-1}$ aprotinin, 20 µg mL$^{-1}$

phenylmethylsulfonyl fluoride)). In total, 1 mg of cell lysate was precleared with 40 µL of Protein A/G Agarose beads for 1 h at 4 °C with agitation. Precleared lysate was incubated with 2 µg Fc or P-selectin-Fc chimera and 20 µL per tube Protein A/G agarose beads at 4 °C overnight with agitation. Precipitated beads were collected and washed with ice-cold lysis buffer. The P-selectin-Fc interacting proteins were eluted with elution buffer (50 mM Tris pH 7.4, 10 mM EDTA, 0.1% Triton, protease inhibitors cocktail) on ice for 5 min. The eluate was collected and denatured by boiling with laemmli sample buffer for 10 min. Interaction of IGF-1R with P-selectin was detected by western blot.

**Immunoprecipitation**. Extracellular protein was extracted with 2 mM EDTA, 150 mM NaCl, 1% NP-40, 50 mM Tris pH 7.4, and protease inhibitor cocktail. Anti-IGF-1R (sc-463, Santa Cruz Biotechnology) was immobilized on cyanogen bromide (CNBr)-activated sepharose (GE Healthcare) according to manufacturer's instruction. CNBr-activated sepharose without coupling antibody was used as a negative control. Cell lysates were precleared with CNBr-activated sepharose for 2 h followed by overnight incubation with anti-IGF-1R or isotype control conjugated beads at 4 °C. The immunoprecipitates were washed with ice-cold lysis buffer and eluted with 0.1% trifluoroacetic acid and neutralized with 1 M Tris pH 8.8. The modification of sLe$^{a/x}$ on IGF-1R was detected via Western blot.

**Histological analysis**. Tissues were fixed in 4% paraformaldehyde, embedded in paraffin, and stained with hematoxylin and eosin. Immunohistochemistry detection was conducted on 5 µm formalin-fixed, paraffin-embedded human omentum tissue section collected from patients with nonmalignant disorders. Heat-induced antigen retrieval was conducted in pH 6.0 sodium citrate buffer. Monoclonal mouse anti-E-selectin, anti-P-selectin, and anti-L-selectin antibodies were used at 1:50 dilution. The streptavidin–biotin immunoperoxidase Histostain SP Bulk kit (Invitrogen) and ImmPACT AEC peroxidase substrate kit (Vector Laboratories) were applied for detection and hematoxylin was used as the nuclear counterstain.

**RNA extraction, reverse transcription, and PCR**. Total RNA was extracted with Trizol and 500 µg of RNA was reverse transcribed to complementary DNA (cDNA) using the first-stranded cDNA synthesis kit (Invitrogen). Specific genes were amplified and quantitated by real-time PCR using primers described in Supplementary Table 3. Real-time PCR was performed using the StepOnePlus real-time detection system and the Power SYBR Green PCR Master Mix (Applied Biosystems). Fluorescent measurements were recorded during each annealing step. The PCR quality and specificity were verified by melting curve analysis and gel electrophoresis. Relative expression was determined by normalizing to the GAPDH endogenous control. These experiments were carried out in duplicate and independently repeated three times.

**FUT5 stable knockdown by shRNA**. Lentiviruses carrying shRNA targeting FUT5 (sequence described in the Supplementary Table 3) or NS shRNA were generated by co-transfection of HEK293 cells with the constructs and lentiviral-packaging plasmids (Sigma-Aldrich) per the manufacturer's instructions. M-CSCs were transduced with the viral particle containing media and cells were selected with 1 µg mL$^{-1}$ puromycin (Calbiochem) 24 h post transduction for 3 days. The knockdown efficiency was verified by q-PCR.

**Statistical analysis**. Results represent mean ± SEM. The significance of differences between categorical variables was determined using the chi-square test. Ordinal variables were assessed using the one-tailed Student's $t$ test.

**Reporting summary**. Further information on research design is available in the Nature Research Reporting Summary linked to this article.

## Data availability
The RNA sequencing data have been deposited in the NCBI Trace Archive under the accession code PRJNA530706. All the other data supporting the findings of this study are available within the article and its supplementary information files or from the corresponding author upon reasonable request.

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

## Acknowledgements

This study was supported by the Hong Kong Research Grant Council grants 17122014. A.S.T.W. is a recipient of the Croucher Senior Fellowship.

## Author contributions

S.-S.L. and C.K.M.I. designed and performed the experiments and analyzed the results. M.Y.H.T. fabricated the microfluidic chip. Y.T. and J.W.Z conducted RNA-seq data analysis. A.A.H established FUT5 stable knockdown cell lines. A.S.C.M. isolated M-/NM-CSCs lines. S.Y. and T.M.C provided primary HPMCs and human omentum specimens. P.P.I. and H.-C.L. provided ovarian tumor specimens. C.L.L. and P.C.N.C. advised and assisted characterization of the glycan. T.O.L. guided the development of double knockout mice. M.K.S.T. helped with the tumor xenograft experiments. J.Z., H.C.S., and A.S.T.W. designed and supervised the study. S.-S.L., C.K.M.I., H.C.S., and A.S.T.W. wrote the manuscript and all authors reviewed the manuscript.

## Additional information

**Competing interests:** The authors declare no competing interests.

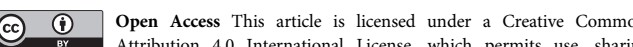

