## [Peer Review File · Nature Communications]

Reviewers' comments:

Reviewer #1 (Expertise: selectins, metastasis, Remarks to the Author):

In this work Li et al studies the potential of selectins to promote ovarian metastasis in the peritoneum. This work shows that metastatic population of cancer stem/tumor-initiating cells (M-CSC) adhere more to the peritoneal mesothelium. While the evidence for selectin involvement in peritoneal adherence of ovarian cells is rather novel, the biology of this mechanism requires to address several points.

Major comments:

1) How the authors came from characterization of metastatic ovarian cancer stem cells to adherence to selectins, which were never before found to be expressed outside of the vasculature remains unclear.

2) Selectin staining in mesothelial cells (Figure 2b) shows if any P-selectin staining E-selectin appears completely negative. To show a de novo expression of selectins in mesothelial cells, real-time quantitative PCR would be "a welcomed" additional evidence. Are really all three selectins expressed on mesothelial cells?

3) The evidence that IGF1-R is really sialyl-Lewis-X carrying ligand on M-CSC is rather indirect. The IP data in Figure 3m-n shows different bands when IGF1-R antibody was used, when compared to HECA-452 Ab positive bands. Please clarify this issue. Again, the IP does not exclude the presence of other ligands carriers on the surface of tumor cells, mediating P-selectin adhesion. Please address this important issue.

4) A classical selectin binding to sLeX ligands is the Calcium dependent. Thus, a good negative control is to test adhesion in the presence of low (5mM EDTA). This is especially relevant, since the Calcium concentration in the interstitium is lower than in the blood circulation. About the Calcium concentration in the peritoneum, there is little evidence.

5) Is the FUT5 detection on IHC really specific??? (Figure 4m). There is only a diffuse cytoplasmic staining at best. Yet, FUT5 would be expected in the Golgi. Finally, the Sigma Aldrich Ab (no Clone number described in the manuscript) is according to the webpage useful just for WB and ELISA). I'm fully aware of the problems using FUT antibodies for IHC. Yet, these points require clarification.

6) Quantification of glycosyltransferases in Figure 4b only in selected M-CSC versus non-metastatic cells, is not sufficient. Quantitative PCR (instead of a gel) should be presented, and more importantly, analysis of other (best patient-derived) cells should be used, for validation of the FUT5 evidence suggested in this work.

Small points:

Description of IGF1-R data in Figure 3m-l are in the text labeled for Figure 4 (page 6 first paragraph), please correct.

Reviewer #2 (Expertise: Ovarian cancer, peritoneal metastasis, Remarks to the Author):

Title: Sialyl LewisX-P-selectin cascade mediates tumor-mesothelial adhesion in ascitic fluid shear flow

Authors: Shan-Shan Li, Carman K. M. Ip, Matthew Y. H. Tang, Yin Tong, et al.

This manuscript deals with an important medical problem, the peritoneal metastasis, which occurs very frequently in high-grade serous ovarian cancer (OvCa). Peritoneal metastasis occurs when tumor cells suspended in ascites adhere to mesothelial cells under shear stress. Despite the strong relationship between metastatic burden and prognosis in OvCa, there are currently no therapies specifically targeting the metastatic process. As stated by the authors, unrevealing the underlying molecular mechanisms regulating this process will conceivably benefit the development of novel treatment strategies. The present work attempts to address this key issue. The authors provide strong evidence that peritoneal dissemination under shear stress is mediated by P-selectin as a key molecule on mesothelial cells and IGF-1R carrying a non-sulfated sialyl-Lewisx (sLex) epitope on metastatic tumor cells. The authors concluded that sLex-P-selectin axis could become a promising therapeutic target.

The results presented are potentially interesting, the experimental approaches are technically sound and the data support the authors conclusions. However, there are some experimental flaws and concerns that should be addressed.

Specific comments:

1.-In this study, the authors employed peritoneal dialysis effluent-derived mesothelial cells to study the expression of selectins and for rolling/adhesion experiments. Throughout the whole manuscript the authors consider mesothelial cells as a stable cell type. However, it is well established that mesothelial cells have enormous plasticity, and that effluent-derived mesothelial cells undergo an epithelial to mesenchymal transition, with a profound genomic reprogramming, which may have also an important role in the initial steps of the peritoneal metastasis (e.g. adhesion). This important issue is not even discussed.

2.- Figure 1 and Figure 2: In the legends to these figures there are some discrepancies between the order of presentation of the panels and the description in the legends.

3.- In general the panels of Figures 1, 2 and 4 are of poor quality and too small, being very difficult to see the details described in the text.

4.- The extended data Figure 5 mentioned "Percentage of adherent SKOV-3 M-CSCs" Is that correct? Or it should say Percentage of adherent HEYA8 M-CSCs?

5.- Novelty: Recently, Carroll MJ, et al. Cancer Res. 2018, have described the role of P-selectin on mesothelial cells in peritoneal carcinomatosis. They demonstrate that tumor cells attached to de novo P-selectin through CD24, resulting in increased tumor cell adhesion in static conditions and rolling under flow.

The authors should discuss to what extent the results presented in this manuscript add new and relevant insight to the understanding of peritoneal metastasis.

Reviewer #3 (Expertise: CSCs, Remarks to the Author):

I think it is difficult to understand the background of the study in this manuscript. The introduction is too short and lacks the information about the importance of cancer stem/tumor-initiating cells. The authors should also describe about the selectins why they focused these proteins. Please discuss more about your findings in discussion.

They used metastatic ovarian cancer cell lines derived from SKOV-2 and HEYA8 cell lines by culturing them in sphere culture conditions. In their previous paper in which they developed these lines, they describe these cells as 'ovarian tumor initiating cells' in *Oncogene*, vol. 32, p2767, 2013. I think they should call them as 'tumor initiating cells' to be consistent. Their findings that the mesothelial dissemination is due to P-selectin-IGF1R interaction in tumor initiating cells are potentially interesting. However, I am sorry that the validation using patient samples is too preliminary. They showed the major results by using only SKOV-2-derived cell line, in *Selp^{-/-}Rag2^{-/-}* mice, for example. As the papers they referred, *Can Res*, vol. 68, p4311, 2008 and *Can Res*, vol. 65, p3025, 2005, I would recommend them to develop such metastatic ovarian cancer cells derived from patient samples and repeat their experiments for validation. They used patient ascites-derived cancer cells in their experiments, however, they need to show that these cancer cells metastasize in their xenograft experiments as shown in Fig. 1c and e.

Specific points:

They used HMPCs. They need to repeat the experiments by using at least several HMPCs derived from different patient samples.

Below we address the reviewers' comments point by point. Page, line, and figure number refer to the revised manuscript.

Reviewer #1

1. How the authors came from characterization of metastatic ovarian cancer stem cells to adherence to selectins, which were never before found to be expressed outside of the vasculature remains unclear.

Reply: The reviewer is correct that there are only a few data exist to support selectins in adhesion outside the vasculature. While such interaction has been extensively studied in blood-borne metastasis, the observation that selectins are expressed on peritoneal mesothelial cells (Ref.#9, 10) suggests that similar receptor/ligand cascade, also initiated by selectin-mediated interaction, may promote homing of tumor cells to the peritoneal mesothelium. However, the detailed molecular mechanisms of this tumor cell lodgement in the peritoneum, particularly under ascitic shear stress, have not been explored. We hypothesize that tumor cells entering the peritoneal microenvironment that express adhesion receptors and ligands specialized to function within ascitic shear stress would have an advantage in establishing residence. This study is therefore to study in detail the receptor on the peritoneal mesothelium, the distinct glycan structure associated with the tumor cells, and finally explore the pathway and elucidate the pathophysiological relevance. Our findings unveil a previously undescribed role for a sLe^x-P-selectin cascade in tumor progression under shear stress of ascitic flow, and provide new perspectives on the sLe^x-P-selectin cascade as a novel drug target against peritoneal metastasis, the most distressing complication (p. 4, lines 19-21; p.20, lines 1-13).

2) Selectin staining in mesothelial cells (Figure 2b) shows if any P-selectin staining E-selectin appears completely negative. To show a de novo expression of selectins in mesothelial cells, real-time quantitative PCR would be "a welcomed" additional evidence. Are really all three selectins expressed on mesothelial cells?

Reply: As suggested, we have conducted real-time quantitative PCR. Our data show that like endothelial cells, used as a reference cell population, selectins are expressed on mesothelial cells, suggesting that their expression outside the vasculature is present, which are consistent with our previous observations and those from other groups (Ref.#9, 10) (Supplementary Fig. 2c) (p. 7, lines 10-14).

3. The evidence that IGF1-R is really sialyl-Lewis-X carrying ligand on M-CSC is rather indirect. The IP data in Figure 3m-n shows different bands when IGF1-R antibody was used, when compared to HECA-452 Ab positive bands. Please clarify this issue. Again, the IP does not exclude the presence of other ligands carriers on the surface of tumor cells, mediating P-selectin adhesion. Please address this important issue.

Reply: The different bands are likely due to use of different antibodies; IGF-1R β vs total IGF-1R (Fig. 5c, d). We have revised the labels to make it clear. The different percent SDS-PAGE gels and the electrophoresis time used which are known to affect relative mobility between two proteins may account for the different bands in Fig. 5d. The reviewer is correct the IP does not exclude the presence of other ligands, and in response to the reviewer's comment, we have tested the known P-selectin ligands bearing sLe^x, such as CD24. We were able to detect CD24; however, unlike IGF-1R, there were no difference in their expression between M-CSCs vs NM-CSCs (Fig. 5f) (p. 10, lines 21-23).

4. A classical selectin binding to sLeX ligands is the Calcium dependent. Thus, a good negative control is to test adhesion in the presence of low (5mM EDTA). This is especially relevant, since

the Calcium concentration in the interstitium is lower than in the blood circulation. About the Calcium concentration in the peritoneum, there is little evidence.

Reply: As suggested, we have evaluated the requirement of calcium by the calcium chelator EDTA, and observed a clear detachment of M-CSCs from P-selectin chimera or HPMCs upon treatment of with EDTA (Supplementary Figure 4). While less than that in serum, there is trace amount of calcium in benign and malignant ascitic fluids (1.5 ± 0.2 mM; Celik et al., Clin Biochem 2002;35:477-481)(p. 8, lines 7-13).

5. Is the FUT5 detection on IHC really specific??? (Figure 4m). There is only a diffuse cytoplasmic staining at best. Yet, FUT5 would be expected in the Golgi. Finally, the Sigma Aldrich Ab (no Clone number described in the manuscript) is according to the webpage useful just for WB and ELISA). I'm fully aware of the problems using FUT antibodies for IHC. Yet, these points require clarification.

Reply: It is correct that there are specificity issues of FUT antibodies. We have deleted the original IHC, and added Fig. 6n showing the successful target gene knockdown of FUT5 from tumor samples harvested from mice at the end of the study by real-time PCR (p.11, lines 18-20).

6. Quantification of glycosyltransferases in Figure 4b only in selected M-CSC versus non-metastatic cells, is not sufficient. Quantitative PCR (instead of a gel) should be presented, and more importantly, analysis of other (best patient-derived) cells should be used, for validation of the FUT5 evidence suggested in this work.

Reply: As suggested, we have evaluated the transcriptional changes by real-time PCR (Fig. 6b, Supplementary Fig. 7a) and observed a consistent and significant increase in expression of B4GalT4, ST3Gal3, ST3Gal4 and FUT5 in M-CSCs, consistent with previous observation in the original manuscript. Similar effect on patients' derived CSCs was also observed (Fig. 6c) (p. 11, lines 3-7).

7. Description of IGF1-R data in Figure 3m-l are in the text labeled for Figure 4 (page 6 first paragraph), please correct.

Reply: Texts have been revised (p. 10, 16-18; p. 35, Fig. 5c, d legends).

Reviewer #2

1. In this study, the authors employed peritoneal dialysis effluent-derived mesothelial cells to study the expression of selectins and for rolling/adhesion experiments. Throughout the whole manuscript the authors consider mesothelial cells as a stable cell type. However, it is well established that mesothelial cells have enormous plasticity, and that effluent-derived mesothelial cells undergo an epithelial to mesenchymal transition, with a profound genomic reprogramming, which may have also an important role in the initial steps of the peritoneal metastasis (e.g. adhesion). This important issue is not even discussed.

Reply: The review is correct that mesothelial cells have enormous plasticity. However, we have limited the use of HPMCs within 3 passages to ensure consistent characteristics over repeated experiments, and the cells retained the epithelial morphology. The cells also did not show change of the expression of mesenchymal (N-cadherin, vimentin) markers (Supplementary Fig. 2) (p.7, lines 3-7).

2. Figure 1 and Figure 2: In the legends to these figures there are some discrepancies between the order of presentation of the panels and the description in the legends.

Reply: The order of presentation of the panels and the description in the legends are revised (p. 33, 34 Fig. 1 and 2 legends).

3. In general the panels of Figures 1, 2 and 4 are of poor quality and too small, being very difficult to see the details described in the text.

Reply: Changes of the panels to higher quality and larger images are made (Fig. 1, 2, 3 and 6).

4. The extended data Figure 5 mentioned “Percentage of adherent SKOV-3 M-CSCs” Is that correct? Or it should say Percentage of adherent HEYA8 M-CSCs?

Reply: The statement “Percentage of adherent SKOV-3 M-CSCs” is correct. Indeed, the figure showed the “percentage of adherent SKOV-3 M-CSC on P-SEL-Fc after RTK inhibitor treatment”. We now labeled the figure to make it clear (Supplementary Fig. 6).

5. Recently, Carroll MJ, et al. Cancer Res. 2018, have described the role of P-selectin on mesothelial cells in peritoneal carcinomatosis. They demonstrate that tumor cells attached to de novo P-selectin through CD24, resulting in increased tumor cell adhesion in static conditions and rolling under flow. The authors should discuss to what extent the results presented in this manuscript add new and relevant insight to the understanding of peritoneal metastasis.

Reply: Although CD24 has been reported to contain sLe^x, there is no evidence shown in that publication. Furthermore, the expression of sLe^x was not proportional to the CD24 expression. While CD24 expression in OVCAR5 is the lowest among all cancer lines tested, the MIP-1 β -activated cell adhesion via P-selectin is higher than lines with high CD24 expression (OVCAR4 and OVCAR8), suggesting that there is other sLe^x-containing ligand(s) of P-selectin. In addition, it is demonstrated that AAM secreted MIP-1 β is required to activate P-selectin expression on mesothelium. While such an effect was demonstrated in culture, it is probable that, in vivo, P-selectin may reach levels that constitutively promote adhesion. Ascites is a proinflammatory milieu rich in macrophages/cytokines. Several aspects of our findings are worth highlighting and may add new and relevant insights to the understanding of peritoneal metastasis: 1) While we have shown CD24 expression in M-CSCs and NM-CSCs, there was no differential expression on CD24, unlike IGF-1R, in M-CSCs and NM-CSCs, suggesting the importance of IGF-1R in mediating the adhesion of M-CSCs. 2) Whereas sulfation is a common feature of P-selectin ligands, our results suggest that the sLe^x-bearing glycan on M-CSCs does not require sulfation for its binding to P-selectin. 3) Our novel sLe^x-bearing P-selectin ligand present on IGF-1R, which is frequently overexpression in ovarian cancer and other peritoneal metastasis models and confers a poor clinical prognosis, mediates IGF-1R independent activation of IGF-1R upon P-selectin binding. 4) We also identified the key glycoenzyme FUT5 being responsible for the ligand expression (p. 10, lines 21-23; p.13, lines 16-23; p. 14, lines 1-2) (Fig. 5f).

Reviewer #3

1. I think it is difficult to understand the background of the study in this manuscript. The introduction is too short and lacks the information about the importance of cancer stem/tumor-initiating cells. The authors should also describe about the selectins why they focused these proteins. Please discuss more about your findings in discussion.

Reply: In reply to the reviewer’s comments, the Introduction is revised to provide more detailed background of the study. We have also revised the Discussion accordingly (p. 4-5 and 11-15).

2. They used metastatic ovarian cancer cell lines derived from SKOV-2 and HEYA8 cell lines by culturing them in sphere culture conditions. In their previous paper in which they developed these lines, they describe these cells as ‘ovarian tumor initiating cells’ in Oncogene, vol. 32, p2767, 2013. I think they should call them as ‘tumor initiating cells’ to be consistent. Their findings that the mesothelial dissemination is due to P-selectin-IGF1R interaction in tumor initiating cells are potentially interesting. However, I am sorry that the validation using patient samples is too preliminary. They showed the major results by using only SKOV-2-derived cell line, in Selp-/- Rag2-/- mice, for example. As the papers they referred, Can Res, vol. 68, p4311, 2008 and Can

Res, vol. 65, p3025, 2005, I would recommend them to develop such metastatic ovarian cancer cells derived from patient samples and repeat their experiments for validation. They used patient ascites-derived cancer cells in their experiments; however, they need to show that these cancer cells metastasize in their xenograft experiments as shown in Fig. 1c and e.

Reply: It is correct that TIC has been used in our earlier work to describe subpopulation of tumor cells possessing the stem cell properties of self-renewal, differentiation and drug resistance, which TIC has been used interchangeably with CSC under these circumstances in the literature. We have used CSC in our later work published in *Oncotarget* 8, 25879-25914, 2017 and *Mol Ther* 26, 70-83, 2017. We therefore did not made any change yet in response to this question. However, if necessary, we could revise the term as suggested. As suggested, patients' ascites derived CSCs (kindly provided by Prof. H.C. Lai who isolated and characterized these cells, Ref.#25) were now included in the experiments as shown in Fig. 1i, Fig 3n and Fig. 6c (p. 6, lines 2-3; p.8, lines 22-23, p.9, lines 1-5; p.11, lines 6-7).

2. They used HMPCs. They need to repeat the experiments by using at least several HMPCs derived from different patient samples.

Reply: Each experiment is now repeated with HPMCs collected from at least 3 different patients. We have now clarified in the Methods section (p. 20, lines 5-6).

REVIEWERS' COMMENTS:

Reviewer #1 (Remarks to the Author):

The raised points were addressed. The story has been improved.

Reviewer #2 (Remarks to the Author):

Title: Sialyl LewisX-P-selectin cascade mediates tumor-mesothelial adhesion in ascitic fluid shear flow

Authors: Shan-Shan Li, Carman K.M. Ip, Matthew Y.H. Tang, et al.

The authors have addressed the major concerns and suggestions of this reviewer. The structure and the scientific impact have been improved significantly in this revised version. This reviewer has no further questions

Manuel Lopez Cabrera

Reviewer #3 (Remarks to the Author):

The reviewer appreciates that they include the experiments using patient samples. However, they do not provide sufficient information. In Fig. 1i, they analyzed 7 ovarian cancer samples, however, they analyzed only sample in Fig. 3n and Fig. 6c. They used several samples and had similar data? Or they performed experiments by using only sample for each experiment? Please clarify this issue. It would be advisable to perform each experiment by using several samples.

Re: NCOMMS-18-09972A "Sialyl LewisX-P-selectin cascade mediates tumor-mesothelial adhesion in ascitic fluid shear flow"

We sincerely thank the reviewer for the helpful comments. In response to Reviewer's comments, we have revised the manuscript that we think further improve the clarity of the manuscript.

Figure numbers refer to the revised manuscript.

Reviewer #3

The reviewer appreciates that they include the experiments using patient samples. However, they do not provide sufficient information. In Fig. 1i, they analyzed 7 ovarian cancer samples, however, they analyzed only sample in Fig. 3n and Fig. 6c. They used several samples and had similar data? Or they performed experiments by using only sample for each experiment? Please clarify this issue. It would be advisable to perform each experiment by using several samples.

Reply: Several samples were used in the experiments and similar results were obtained using different patient samples. We had analyzed three ovarian cancer samples in 7 mice and two patient samples in 6 mice in Fig. 3n. In Fig. 6c, we had analyzed four ovarian cancer samples and two samples from health donor. The sample number used is now included in the figure legend.